# Phospholipid Scramblase 4 (PLSCR4) Regulates Adipocyte Differentiation via PIP3-Mediated AKT Activation

**DOI:** 10.3390/ijms23179787

**Published:** 2022-08-29

**Authors:** Lisa A. G. Barth, Michèle Nebe, Hermann Kalwa, Akhil Velluva, Stephanie Kehr, Florentien Kolbig, Patricia Prabutzki, Wieland Kiess, Diana Le Duc, Antje Garten, Anna S. Kirstein

**Affiliations:** 1University Hospital for Children & Adolescents, Center for Pediatric Research, Leipzig University, 04103 Leipzig, Germany; 2Institute of Pharmacology, Pharmacy and Toxicology, Leipzig University, 04107 Leipzig, Germany; 3Institute of Human Genetics, Leipzig University Medical Center, 04103 Leipzig, Germany; 4Department of Evolutionary Genetics, Max Planck Institute for Evolutionary Anthropology, 04103 Leipzig, Germany; 5Bioinformatics Group, Department of Computer Science, Interdisciplinary Center for Bioinformatics, Leipzig University, 04107 Leipzig, Germany; 6Institute for Medical Physics and Biophysics, Leipzig University, 04107 Leipzig, Germany

**Keywords:** PLSCR4, PTEN, lipoma, scramblase, PIP3, adipogenesis

## Abstract

Phospholipid scramblase 4 (PLSCR4) is a member of a conserved enzyme family with high relevance for the remodeling of phospholipid distribution in the plasma membrane and the regulation of cellular signaling. While PLSCR1 and -3 are involved in the regulation of adipose-tissue expansion, the role of PLSCR4 is so far unknown. PLSCR4 is significantly downregulated in an adipose-progenitor-cell model of deficiency for phosphatase and tensin homolog (PTEN). PTEN acts as a tumor suppressor and antagonist of the growth and survival signaling phosphoinositide 3-kinase (PI3K)/AKT cascade by dephosphorylating phosphatidylinositol-3,4,5-trisphosphate (PIP3). Patients with PTEN germline deletion frequently develop lipomas. The underlying mechanism for this aberrant adipose-tissue growth is incompletely understood. PLSCR4 is most highly expressed in human adipose tissue, compared with other phospholipid scramblases, suggesting a specific role of PLSCR4 in adipose-tissue biology. In cell and mouse models of lipid accumulation, we found PLSCR4 to be downregulated. We observed increased adipogenesis in PLSCR4-knockdown adipose progenitor cells, while PLSCR4 overexpression attenuated lipid accumulation. PLSCR4 knockdown was associated with increased PIP3 levels and the activation of AKT. Our results indicated that PLSCR4 is a regulator of PI3K/AKT signaling and adipogenesis and may play a role in PTEN-associated adipose-tissue overgrowth and lipoma formation.

## 1. Introduction

Phosphatase and tensin homolog (PTEN) acts an important tumor suppressor and a negative regulator of cell growth and survival signaling by antagonizing the phosphoinositide 3-kinase (PI3K)/AKT/mammalian target of rapamycin (mTOR) cascade. PI3K is activated through a plethora of hormones and growth factors, including insulin [1]. The activated kinase then phosphorylates phosphatidylinositol-4,5-bisphosphate (PIP2) to phosphatidylinositol-3,4,5-trisphosphate (PIP3), which is a second messenger essential for the activation of AKT via phosphorylation. AKT, as a PI3K downstream target, is a key regulator of cell growth, survival, and signaling. PTEN antagonizes PI3K action by dephosphorylating PIP3 to PIP2. A deficiency in functional PTEN due to heterozygous germline mutations leads to a constant activation of PI3K/AKT/mTOR signaling, causing a number of disorders, which are summarized as PTEN hamartoma tumor syndrome (PHTS) [2]. The wide clinical spectrum of this disease includes an increased risk of malignant and benign tumors. The formation of adipose-tissue tumors known as lipomas is observed in up to 39% of PHTS patients [3]. We previously reported on a pediatric PHTS patient with severe lipomatosis [4]. Although mTOR inhibitor rapamycin improves outcomes in patients with PHTS, there is to date no sufficient pharmacological therapy, and the underlying mechanism of lipoma formation is incompletely understood.

To investigate the mechanisms that lead to aberrant adipose-tissue growth in PHTS, we previously established a cell model of PTEN deficiency in adipose progenitor cells (APCs). In the PTEN-knockdown APCs, we observed increased differentiation and proliferation compared with control cells. Via RNA sequencing, we identified 1379 differently expressed genes when comparing PTEN-knockdown with control APCs [5]. To narrow down potential candidates involved in the regulation of lipid accumulation, we compared our results with gene expression datasets from two other models for fat accumulation and distribution [6,7]. One of the genes significantly regulated in all three datasets was phospholipid scramblase 4 (PLSCR4), which is one out of five members of the phospholipid scramblase family that occur in humans. Phospholipid scramblases are plasma membrane proteins that translocate phospholipids between the two monolayers of the membrane lipid bilayer [8]. This process is important for many cell signaling functions, such as cell fusion, apoptosis, and blood coagulation [9]. Contrary to other transmembrane phospholipid transporters (flippases and floppases) scramblases are non-selective and ATP-independent bidirectional transporters. Their enzyme activity depends on cellular calcium concentrations [10]. The five human phospholipid scramblases exhibit distinct tissue specificity and occur in different cell types. PLSCR1 is predominantly and highly expressed in the heart, liver, kidney, and pancreas and plays an important role in blood cells such as platelets and erythrocytes. PLSCR2 is exclusively expressed in testis. PLSCR3 is highly expressed in muscles, the endometrium, and adipose tissue. PLSCR3 and, to a lesser extent, PLSCR1 knockout mice showed an increase in abdominal fat mass and in the size of adipocytes compared with wild-type mice. This aberrant accumulation of abdominal fat was accompanied by insulin resistance, glucose intolerance, and dyslipidemia [11]. PLSCR5 is associated with brain function [12]. PLSCR4 was shown to be regulated in models of lipopolysaccharide-induced acute respiratory distress syndrome [13], in sickle cell disease [14], and lipid and cholesterol metabolism [15]. However, little is known about PLSCR4 function in general and specifically in adipocytes.

Given the known effect of PLSCR3 knockout on adipose-tissue expansion and the fact that PLSCR4 is significantly downregulated in PTEN-knockdown adipose progenitor cells, we propose that PLSCR4 may play a role in lipid accumulation and adipocyte development.

## 2. Results

### 2.1. PLSCR4 Is Closely Related to PLSCR1 and Highly Expressed in Adipose Tissue

To identify functional similarities among PLSCR family members, we analyzed the evolutionary conservation in human, chimp, and mouse, based on amino-acid-sequence distance among all PLSCRs. We found that PLSCR4 is the most closely related to PLSCR1 and PLSCR2 in all considered species (Appendix A). All PLSCRs except for PLSCR3 are localized on chromosome 3, adjacent to each other. This, together with the evolutionary distance, may suggest that PLSCR4 arose due to a duplication of PLSCR1. We analyzed the PLSCR4 sequence across all available mammalians and identified that monotremes lack both PLSCR2 and PLSCR4 (Figure 1a). The maintained synteny of genes adjacent to PLSCR2 and PLSCR4 in these species (Figure 1b) supports the fact that PLSCR4, and most probably also PLSCR2, arose through a duplication of PLSCR1 at the split between monotremes and marsupials, which occurred more than 100 million years ago [16]. While PLSCR4 does not seem to be under selection, PLSCR1 shows strong selection in the whole mammalian branch [17], which may imply that PLSCR4 gene duplication was favorable for functional redundancy to substitute PLSCR1.

Using PTEE [18], a tool we developed based on data from GTEx [19], we found that *PLSCR4* is the most highly expressed in human subcutaneous and visceral adipose tissue compared with all other phospholipid scramblases (Figure 1c), suggesting a specific role of *PLSCR4* in adipose-tissue biology.

### 2.2. PLSCR4 Is Downregulated in PTEN-Knockdown Adipocyte Progenitor Cells and Localized on the Plasma Membrane

Based on our results from comparative genomics analyses, we set out to functionally characterize PLSCR4 in APCs. To verify RNA sequencing results, we performed small interfering (si)-RNA-mediated PTEN knockdown (by 90.2% ± 0.5% at the protein level (Figure 2a,b and Appendix A) and by 79.5% ± 13.9% at the mRNA level (Figure 2d); both *p* < 0.05) in adipocyte progenitor cultures from three different adipose-tissue donors. PLSCR4 expression was significantly reduced at the protein level (by 52.5% ± 6.5%; *p* < 0.05) (Figure 2a,c and Appendix A) and at the mRNA level (by 56.6% ± 10.8%; *p* < 0.05) (Figure 2e). PLSCR4 appeared to be localized on the plasma membrane and in the cytosol but not in the nucleus when detected via immunofluorescence (Figure 2f).

### 2.3. PLSCR4 Expression Is Altered in Different Models of Lipid Accumulation

To analyze and investigate the role of PLSCR4 in the regulation of adipose-tissue distribution and adipogenesis, we used additional models of lipid accumulation. The first was a lipoma-bearing mouse model with a conditional double knockout (DKO) of the *Pten* and *Retinoblastoma* genes (*Pten*/*Rb* DKO) in osteoprogenitor cells [20]. We compared *Plscr4* gene expression in lipoma tissue from *Pten*/*Rb* DKO with inguinal (ing) and epididymal (epi) white adipose tissue (WAT) from *Cre* negative-control mice. We observed decreases in *Plscr4* expression of 67.7% ± 15.4% in lipoma *Pten*/*Rb* DKO tissue compared with epiWAT (*n* = 7; *p* = 0.0019) and of 58.4% ± 23.5% compared with ingWAT (*n* = 7; *p* = 0.0318) (Figure 3a).

We used an APC strain termed LipPD1 established from the lipoma tissue of a patient with a germline mutation in the PTEN gene [4,21] and APCs from obese but otherwise healthy individuals to investigate PLSCR4 expression during adipocyte differentiation. In these adipocyte models, we observed a steady decline in PLSCR4 expression during adipogenesis at the mRNA level (Figure 3b). In LipPD1 cells, PLSCR4 expression declined significantly after 4, 8, and 12 days (*p* < 0.01). The same was observed at the protein level in APCs and LipPD1 cells during adipocyte differentiation (Figure 3c and Appendix A).

### 2.4. Knockdown and Overexpression of PLSCR4 Affect Differentiation of Adipocyte Progenitor Cells

To investigate a potential functional role of PLSCR4 in adipocyte development and lipoma formation, we knocked down or overexpressed PLSCR4 in APCs and examined the changes in proliferation and adipocyte differentiation compared with controls. We achieved a nearly complete PLSCR4 knockdown (by 95.8% ± 15.0% at the protein level (Figure 4a and Appendix A) and by 94.0% ± 29.9% at the mRNA level (Appendix A)) but did not see a significant effect on the proliferation of APCs (Appendix A). In contrast, PLSCR4-knockdown APCs showed a significant increase in lipid accumulation after in vitro differentiation. The differentiated cell fraction in PLSCR4-knockdown cells was increased 1.37 ± 0.09-fold (*n* = 7, *p* = 0.0056) compared with control APCs (Figure 4c,d).

To confirm these results, PLSCR4 was overexpressed in APCs (8.2 ± 1.8-fold compared with cells transfected with a control plasmid; *n* = 3; *p* = 0.0446) (Figure 4b and Appendix A). In line with the result obtained in PLSCR4 KD cells, increased PLSCR4 protein expression led to a reduction in the percentage of adipocytes (by 25.3% ± 7.3%; *n* = 3; *p* = 0.0741) (Figure 4e,f). We observed no significant effects on proliferation in PLSCR4-overexpressing APCs (Appendix A).

### 2.5. AKT Phosphorylation and PIP3 Immunofluorescence Staining Are Increased in PLSCR4 KD APCs

Since Plscr3 knockout in mice was shown to influence insulin signaling [11], we analyzed AKT activation in PLSCR4 KD APCs compared with control cells via Western blot. We saw a significant increase in AKT phosphorylation in PLSCR4 KD APCs of 52.0% ± 13.6% compared with control cells (*n* = 5; *p* = 0.0186) (Figure 5a,b).

To test whether PLSCR4 elicited effects similar to those of PLSCR3 on the expression of the components of the insulin receptor signaling pathway, we determined the expression of insulin receptor β subunit (IR β), insulin receptor substrate 1 (IRS-1), and PTEN. Neither IR β, IRS-1, nor PTEN expression was regulated in PLSCR4 KD cells at the protein level (*n* = 5) (Figure 5a,c and Appendix A).

We then checked if PLSCR4, as phospholipid scramblase, affected cellular PIP3 levels, which could lead to an increased phosphorylation of AKT. We determined the level of PIP3 in PLSCR4 KD and control APCs via immunofluorescence staining and detected 1.92 ± 0.21-fold higher PIP3 levels in PLSCR4 KD APCs (*n* = 4; *p* = 0.0226) (Figure 5d,e), corresponding to increased AKT phosphorylation. In contrast, PIP2 levels, as determined via high-performance liquid chromatography–mass spectrometry (HPLC-MS) measurements, were reduced by 23.1 ± 4.7% in PLSCR4 KD cells (*n* = 4; *p* = 0.0215) (Appendix A). We further investigated whether the phosphorylation of AKT led to an increased expression of lipogenic transcription factor sterol regulatory element-binding protein 1 (SREBP1), which is known to be transcriptionally regulated by AKT target FOXO1. The SREBP1 protein was elevated in PLSCR4 KD cells by 28.9% ± 11.6% compared with control cells (*n* = 3; *p* = 0.116) (Appendix A).

## 3. Discussion

According to a recent study, 39% of patients with PTEN germline mutations develop lipomas [3]. The mechanism underlying this adipose-tissue dysfunction is incompletely understood. In this study, we analyzed the role of a potentially involved factor, PLSCR4, in adipose tissue and its impact on lipoma formation in PHTS patients. Comparative genomics indicated that both PLSCR2 and PLSCR4 appear to have arisen through a duplication of PLSCR1 at the split of the basal mammalian lineages (monotremes and marsupials do not have PLSCR4). The tissue distribution of PLSCRs may be related to the function they have, either being specialized in one tissue or assuring functional redundancy for PLSCR1. The fact that PLSCR4 is twice as much expressed in subcutaneous adipose tissue compared with PLSCR1 and PLSCR3 may suggest a high relevance of adipogenesis. This is a vital metabolic pathway, and genetic redundancy is necessary, which may explain why these genes have been maintained during evolution over the past 100 million years.

RNA sequencing of an adipocellular model of PTEN deficiency showed that *PLSCR4* was significantly downregulated in PTEN KD APCs [5]. We confirmed these results by showing the downregulation of PLSCR4 in PTEN KD APCs at the mRNA and protein levels. As previously described for PHTS lipoma cells [21] and PTEN KD APCs, the downregulation of PTEN led to enhanced proliferation and adipogenesis [5]. To investigate whether a downregulation of PLSCR4 contributed to these changes, we performed PLSCR4 KD and overexpression experiments in APCs. Altering PLSCR4 expression had no influence on adipose progenitor proliferation, but the differentiated cell fraction in PLSCR4 KD cells was increased in comparison with control cells. The opposite effect was observed in PLSCR4 overexpression experiments, indicating a role of PLSCR4 in regulating adipogenesis. Our results matched the mouse phenotype described by Wiedmer et al., who previously showed that *Plscr1* or *Plscr3* knockout led to adipose-tissue expansion in mice [11]. The authors reported a decreased expression in insulin receptor (IR-β) and insulin receptor substrate (IRS-1), leading to insulin resistance. Therefore, we measured IR-β and IRS-1 protein levels in PLSCR4 KD APCs but could not observe any changes. Instead of detecting a decreased activation of insulin signaling, we observed an increased activation of downstream signaling component AKT. This finding supported the observed increase in adipogenesis. Since we detected no changes in upstream pathway components, we analyzed cellular PIP3 levels as a potential cause of increased AKT phosphorylation. We observed increased PIP3, but decreased PIP2, in PLSCR4 KD APCs, which was in line with the observed AKT activation and adipogenesis. This observation indicated a direct or indirect effect of PLSCR4 on cellular PIP3 levels. Further studies should aim at deciphering the nature of this interplay and the exact role of PLSCR4 in the regulation of PIP second messengers. Additionally, we observed an increased expression of lipogenic transcription factor SREBP1 in PLSCR4 KD cells, which is a downstream target of the PI3K pathway. Elevated SREBP1 levels may induce the observed effect on adipogenesis [5].

PLSCR1 contains a nuclear localization sequence and was found to be localized both on the plasma membrane and in the nucleus [22]. To investigate the cellular localization of PLSCR4, we performed PLSCR4 immunofluorescence staining, demonstrating the membrane localization but no nuclear localization of PLSCR4 in APCs [23]. Since Plscr3 knockout mice showed a higher accumulation of abdominal fat as well as insulin resistance, glucose intolerance, and dyslipidemia [11], we aimed to investigate the role of PLSCR4 in other models of lipid accumulation. We observed that PLSCR4 expression was downregulated in lipoma tissue from conditional *Pten*/*Rb* double knockout mice in comparison with epididymal and inguinal WAT of control mice. Similarly, during the adipogenesis of lipoma cells from a patient with PHTS, we observed a downregulation of PLSCR4 at the mRNA and protein levels, suggesting an association of reduced PLSCR4 with adipocyte differentiation.

The present study was designed to investigate the role of PLSCR4 in adipose progenitor proliferation and differentiation in order to better understand the molecular causes of lipoma development in patients with PHTS. We found that PLSCR4 downregulation led to increased cellular PIP3 levels and AKT activation and an enhanced adipose differentiation of APCs. Since we did not observe nuclear localization, we propose a direct effect of membrane-associated PLSCR4 on cellular PIP3 levels, but further research is needed to elucidate the mechanisms leading to these alterations. In lipoma tissue from a mouse model of PTEN deficiency, we observed a lower expression of PLSCR4 compared with control adipose tissue, proposing a contribution of PLSCR4 downregulation to the adipose-tissue overgrowth observed in patients with PHTS.

## 4. Materials and Methods

### 4.1. PLSCR Gene Synteny and Expression Patterns

To identify the evolutionary distance among the members of the PLSCR family, we performed multiple-sequence alignment of the protein sequence in three (human, mouse, and chimpanzee) species using Muscle aligner (Version 5) [24] and computed a distance matrix. We further built evolutionary trees for all PLSCRs in all mammals after retrieving the sequences from the NCBI repository. For monotremes in which we could not identify PLSCR4, an outgroup basal species, and the species for which we calculated the distance matrix, we performed synteny analyses including adjacent genes. We plotted the syntenic regions using R package genoPlotR [25].

### 4.2. Cell Culture and In Vitro Adipocyte Differentiation

We used human primary adipose progenitor cells (APCs) isolated from the stromal vascular fraction (SVF) of adipose tissue, obtained from visceral adipose tissue of healthy donors resected during bariatric surgery of obese (BMI > 30 kg/m^2^), non-diabetic adult patients under 40 years of age. For each experiment, we used cell cultures from at least 3 different donors. Moreover, we used LipPD1, which were isolated from lipoma tissue that was resected for diagnostic and therapeutic reasons from a pediatric patient with PHTS [4,21]. Cells were maintained in Dulbecco’s modified Eagle’s medium (DMEM)/F12 medium supplemented with 10% fetal calf serum, glutamine (2 mM), biotin (33 mM), and pantothenic acid (17 mM) at 37 °C in a humidified atmosphere containing 5% CO_2_.

For adipose differentiation, 120,000 cells/12 wells were plated with culture medium. The medium was changed to differentiation medium 24 h later (day 0) (Dulbecco’s modified Eagle’s medium/F12 containing 8 mg/mL D-biotin, 10 mg/mL D-pantothenic acid, 2 μM rosiglitazone, 25 nM dexamethasone, 0.5 mM methylisobuthylxantine, 0.1 μM cortisol, 0.01 mg/mL apotransferrin, 0.2 nM triiodotyronin, and 20 nM human insulin (45)), and cells were kept in the differentiation medium for 4, 8, or 12 days.

### 4.3. Quantitative Real-Time PCR (qPCR)

RNA isolation from mouse lipoma tissue, and epididymal and inguinal white adipose tissue was performed using TRIzol Reagent (Thermo Fisher Scientific, Waltham, MA, USA); the samples were homogenized using TissueLyser II (Qiagen, Hilden, Germany). RNA from cell cultures was extracted using RNeasy Mini Kit (Qiagen). Reverse transcription and qPCRs were performed as previously described [21]. Appendix A contains the list of primers used for qPCR assays. Results were normalized to housekeeping genes hypoxanthine phosphoribosyltransferase and TATA box–binding protein. We performed SYBR green assays using Takyon Low Rox SYBR MasterMix dTTP Blue (Eurogentec).

### 4.4. PLSCR4 Expression Analysis in Mouse Lipoma and White Adipose Tissue

Lipoma and white adipose-tissue samples were taken from female Osx-Cre Pten^lox/lox^ Rb^lox/lox^ mice (*n* = 7) and Cre negative-control mice (*n* = 7). The phenotype and the breeding of these mice from Osx1-GFP::Cre mice, Rb1^lox/lox^ mice, and Pten^lox/lox^ mice as well as the associated PCR genotyping protocols were previously described [20]. Sacrifice and organ collection were carried out according to guidelines approved by the local authorities of the State of Saxony, Germany, as recommended by the responsible local animal ethics review board (Regierungspräsidium Leipzig, Germany; TVV30/19).

### 4.5. PLSCR4 and PTEN siRNA Knockdown

Twenty-four hours before transfection, APCs were seeded at a density of 1400 cells/cm^2^ for optimal growing conditions. Transfection was performed using Neon Transfection System 100 µL Kit (Invitrogen; Thermo Fisher Scientific, Inc., Waltham, MA, USA). Before transfection, cells were harvested via trypsination and washed with DPBS. For transfection, cell pellets were mixed with *PLSCR4* siRNA (PLSCR4 Silencer^®^ Select siRNA s32644; Ambion, Thermo Fisher Scientific, Inc.), two *PTEN* siRNAs (s325 and s326; both Ambion) or control siRNA (Silencer^®^ Negative Control No. 1 siRNA; Ambion, Thermo Fisher Scientific) to a final concentration of 10 nM in culture medium after transfection. Cell pellets were resuspended in 110 µL R-buffer for transfection via electroporation in Neon 100 µL tips at 1300 V, 2 pulses, and 20 ms. Afterwards, cells were transferred to prewarmed medium and seeded for functional assays as described below. A change in medium was performed after 24 h.

### 4.6. PLSCR4 Overexpression Mediated by Plasmids

One day prior to transfection, SVF cells were plated at a density of 1400 cells/cm^2^ for optimal growing conditions. The transfection was performed using Neon Transfection System 100 µL Kit (Invitrogen; Thermo Fisher Scientific, Inc.). Before transfection, cells were harvested via trypsination and washed with DPBS.

For PLSCR4 overexpression, 2 µg of PLSCR4 plasmid (PLSCR4 pcDNA3.1+/C-(K)DYK #OHu03299C; GeneScript) or control plasmid (c-Flag pcDNA3; gift from Stephen Smale; Addgene plasmid #20011; http://n2t.net/addgene:20011 (accessed on 26 August 2022) [26]) was added to the cell pellets, which were resuspended in 110 µL R-buffer for transfection via electroporation in Neon 100 µL tips at 1300 V, 2 pulses, and 20 ms. Afterwards, cells were transferred to prewarmed medium and seeded for functional assays as described below. A change in medium was performed after 24 h.

### 4.7. Proliferation Assays

After knockdown via siRNA or overexpression (OE) via plasmid, cells were functionally characterized. To examine the proliferation, we seeded cells at a density of 2000 cells/well in a 96-well plate. PLSCR4 KD and PLSCR4 OE cells in comparison with control cells were incubated in 100 µL/well medium, and the medium was replaced every 2–3 days. Cells were fixed on day 1 and day 7. Nuclei of fixed cells were stained with Hoechst 33342 (1 µg/mL; Sigma-Aldrich, St. Louis, MO, USA), and fluorescence was measured using a plate scanner.

### 4.8. Adipogenesis Assays

For adipogenesis assays, cells were seeded in culture medium at a density of 15,000 cells/well in a 96-well plate. After 24 h, the medium was replaced with differentiation medium (DMEM/F12 containing 2 µmol/L rosiglitazone, 25 nmol/L dexamethasone, 0.5 mmol/L methylisobuthylxantine, 0.1 µmol/L cortisol, 0.01 mg/mL apotransferrin, 0.2 nmol/L triiodotyronin, and 20 nmol/L human insulin) and incubated at 37 °C for 8 days. After 8 days, cells were fixed in 4% paraformaldehyde and washed with DPBS. Nuclei were stained with Hoechst 33342 (1 µg/mL; Sigma), and lipids were stained with fluorescent dye Nile Red (0.5 µg/mL; Sigma) for 10 min in DPBS in the dark. After washing, microscope images were taken using EVOS FL Auto 2 Cell Imaging System (Invitrogen; Fisher Scientific). Cell count (Hoechst) and differentiated cell count (Nile Red) were determined using Celleste Image Analysis Software (Thermo Fisher Scientific).

### 4.9. Immunofluorescence Staining

For immunofluorescence staining, 30,000 cells were seeded on round 25 mm glass coverslips in 6-well plates. After 24 h, cells were fixed with 4% paraformaldehyde, and permeabilized and blocked in IF buffer (5% BSA + 0.3% Tween in DPBS) for 1 h at room temperature (RT). Cells were stained with anti-PLSCR4 antibody (ab233005; Abcam; 1:100 in IF buffer), Ki-67 antibody (P0447; Dako; 1:200 in IF buffer), or anti-PIP3 antibody (Purified Anti-PtdIns(3,4,5)P3 IgM; Z-P345 Echelon Biosciences; 1:200 in IF buffer) overnight at 4 °C. After washing with IF buffer, cells were incubated with fluorescently labeled secondary antibodies (1:1000 in IF buffer) for 1 h at RT in the dark (for PLSCR4, Goat anti-Rabbit IgG (H+L) Cross-Adsorbed Secondary Antibody (Alexa Fluor™ 488); for Ki-67, Alexa Flour 488 goat anti-Mouse IgM; for PIP3, Goat anti-Mouse IgM (Heavy chain) Cross-Adsorbed Secondary Antibody (Alexa Fluor 488); all Invitrogen). Cells were washed with IF buffer and DPBS and mounted on glass slides overnight at RT using ProLong™ Diamond Antifade Mountant (Thermo Fisher Scientific). For imaging, a confocal microscope, Leica TCS SP8, and LAS X Software (Leica) were used. Ten cells were selected for each condition. Corrected total cell fluorescence (CTCF) for PIP3 immunofluorescence staining was determined with ImageJ [27] using the following calculation: CTCF = Integrated Density − (Area of selected cell × Mean fluorescence of background readings) [28].

### 4.10. Western Blot Analysis

After transfection of siRNA or plasmids, cells were seeded in 6-well plates with a density of 60,000 cells/well in culture medium. After 24 h, the medium was replaced with serum-free medium. The next day, cells were harvested via trypsination and frozen at −80 °C as pellets. For Western blot analysis, proteins were extracted, and immunoblotting was performed as previously described [4]. In brief, we used 10 µg of protein per lane and separated the proteins in polyacrylamide gels via electrophoresis. After semi-dry blotting, nitrocellulose membranes were incubated with primary and secondary antibodies according to Appendix A. Images of all Western blots are provided in Appendix A.

### 4.11. PIP2/PIP3 Quantification via High-Performance Liquid Chromatography–Mass Spectrometry (HPLC-MS)

Pellets from 106 cells were thawed on ice, and 0.23 nmol of phosphatidylinositol 3,4-bisphosphate (17:0-20:4 PI (3,4)P2) and 0.21 nmol phosphatidylinositol 3,4,5-trisphosphate (17:0-20:4 PI (3,4,5)P3) (Avanti Polar Lipids, Alabaster, AL, USA) were added to each sample before homogenization. Lipid extraction and derivatization were performed according to Clark et al. [29] with slight adaptations. In brief, 750 µL of ice-cold MeOH/CHCl_3_/1 M HCl ((20/10/1, *v*/*v*/*v*)) was added to each sample, mixed thoroughly, and incubated on ice for 5 min. Subsequently, 725 µL of CHCl3 and 170 µL of 2 M HCl were added, mixed thoroughly, and incubated on ice for 10 min, followed by centrifugation to achieve phase separation (4 °C, 5 min, 2000× *g*). The lower phase was collected and re-extracted with the addition of 708 µL of MeOH/CHCl3/0.01 M HCl ((2/1/0.375, *v*/*v*/*v*)). Again, the lower phase was collected, and 200 µL of MeOH was added for derivatization. Derivatization was performed using 50 µL of 2 M trimethylsilyldiazomethane in hexane. The components were allowed to react for 10 min at RT, before the reaction was stopped with the addition of 6 µL of glacial acetic acid. To induce phase separation, 700 µL of MeOH/CHCl_3_/H_2_O ((2/1/0.375, *v*/*v*/*v*) was added to the mixture and mixed thoroughly (4 °C, 5 min, 2000× *g*). The lower phase was collected and dried in vacuo. Extracts were resuspended in 25 µL of isopropanol and briefly sonicated. A volume of 25 µL of acetonitrile was added prior to chromatographic separation. The injection volume was 20 µL. Lipid extracts were separated using reversed phase liquid chromatography on a Ultimate 3000 RSLCnano system (Thermo Fisher Scientific, Dreieich, Germany) equipped with a Kinetex C18 column (Phenomenex; 2.6 µm 2.1 × 100 mm; 100 Å) via isocratic elution with solvent A (H_2_O containing 5 mM NH_4_HCO_3_ and 0.1% (*v*/*v*) formic acid) and B (acetonitrile/isopropanol, 50:50 (*v*/*v*) containing 0.1% (*v*/*v*) formic acid). Separation was performed at 35 °C at a flow rate of 0.75 mL/min using a constant flow of 90% B. Mass spectra were recorded on an AB SCIEX QTRAP 5500 mass spectrometer (Sciex, Framingham, MA, USA) equipped with an ESI probe. Mass spectra were acquired in positive multiple reaction monitoring (MRM) mode. MS parameters were set as follows: spray voltage, 5.5 kV (ESI positive mode); temp. 350 °C; curtain gas, 42.0; collision gas, 8.0; ion source gas, 30.0; entrance potential, 10 V; declustering potential, 281 V for PIP2; collision energy, 37 V; excitation energy, 6 V for PIP2. Quantification was based on the determination of area-under-the-curve values and normalized to the protein content of each sample.

### 4.12. Statistical Analysis

Experiments were independently performed ≥ 3 times, and results were statistically analyzed using GraphPad Prism 9 software (GraphPad Software, Inc., San Diego, CA, USA). Results from unpaired replicate experiments were presented as means ± SEM. For the comparison of two conditions in paired experiments (e.g., control vs. respective KD), we analyzed the results using paired Student’s *t*-test. For comparison of multiple groups, we used a one-way analysis of variance (ANOVA) followed by Dunnett´s multiple comparisons test.

## Figures and Tables

**Figure 1 ijms-23-09787-f001:**
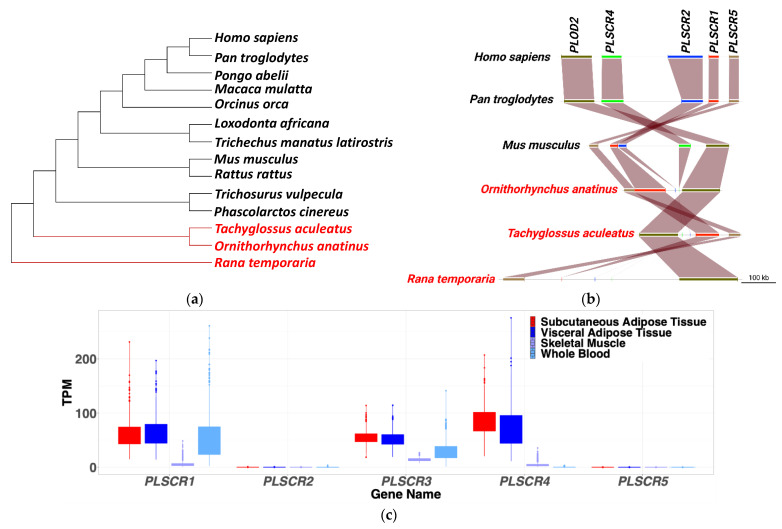
*PLSCR* genes synteny and expression patterns. (**a**) Phylogenetic tree of mammalian representatives showing the absence (labeled red) of *PLSCR4* in Monotremata and the basal outgroup representative of amphibians (*Rana temporaria*). (**b**) Gene synteny of *PLSCR* genes including the species lacking PLSCR4 (labeled in red). *PLOD2* was included because it is the neighboring gene of PLSCR4. (**c**) Expression profiles of *PLSCR* genes in different human tissues based on TPMs (Transcripts Per Million) obtained from the GTEx database. PLSCR4 is highly expressed in adipose tissue. PLSCR, phospholipid scramblase; *PLOD2, Procollagen-Lysine,2-Oxoglutarate 5-Dioxygenase 2*.

**Figure 2 ijms-23-09787-f002:**
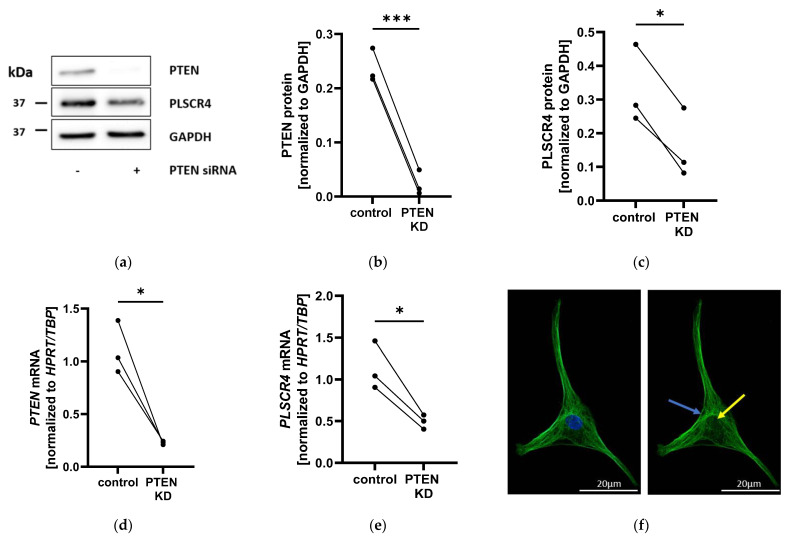
PLSCR4 was downregulated in PTEN knockdown (PTEN KD) adipocyte progenitor cells (APCs). (**a**) Western blots of PTEN KD APCs: One representative image out of three of PTEN, PLSCR4, and GAPDH protein detection is shown. (**b**) Densitometric analysis of Western blots: PTEN protein was reduced by 90.2% ± 0.5% (normalized to GAPDH; *n* = 3; *p* = 0.0006). (**c**) In PTEN KD APCs, PLSCR4 was downregulated by 52.5% ± 6.5% (normalized to GAPDH; *n* = 3; *p* = 0.0151). (**d**) PTEN and (**e**) *PLSCR4* gene expression of PTEN KD APCs: *PTEN* was reduced by 79.5 % ± 13.9 % (normalized to *HPRT* and *TBP*; *n* = 3; *p* = 0.0291). *PLSCR4* was reduced by 56.6% ± 10.8 % (normalized to *HPRT* and *TBP*; *n* = 3; *p* = 0.0343) in PTEN KD cells. (**f**) Hoechst nuclear staining (blue) and PLSCR4 immunofluorescence staining (green) in adipocyte progenitor cells: PLSCR4 immunofluorescence staining indicated plasma membrane localization (blue arrow), while the nuclear area was not stained (yellow arrow). Matched results were visualized via lines between data points (control versus PTEN KD). *p*-values were determined via paired *t*-test (* *p* < 0.05, *** *p* < 0.001). PTEN, phosphatase and tensin homolog; PLSCR4, phospholipid scramblase; HPRT, hypoxanthine phosphoribosyl transferase; TBP, TATA-box binding protein; GAPDH, glycerinaldehyde-3-phosphate dehydrogenase.

**Figure 3 ijms-23-09787-f003:**
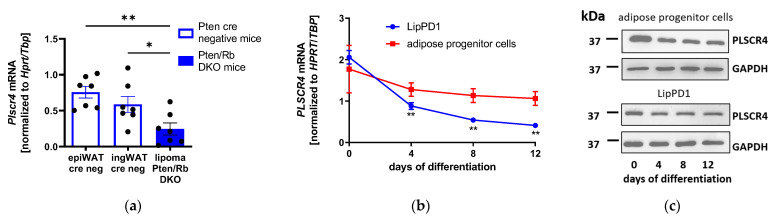
PLSCR4 expression in different models of lipid accumulation. (**a**) *Plscr4* expression in lipoma tissue from lipoma *Pten*/*Retinoblastoma* (Rb) double knockout (DKO) mice in comparison with epididymal WAT (epiWAT cre neg) and inguinal WAT (ingWAT cre neg) from *Cre* negative control mice: *Plscr4* expression was diminished by 67.7% ± 15.4% in lipoma *Pten*/*Rb* DKO compared with epiWAT (normalized to *Hprt* and *Tbp*; *n* = 7; *p* = 0.0019) and by 58.4% ± 23.5% (normalized to *Hprt* and *Tbp*; *n* = 7; *p* = 0.0318) compared with ingWAT. (**b**) *PLSCR4* expression during adipogenesis in human PTEN-deficient lipoma cells (LipPD1) and adipocyte progenitor cells (APCs) at mRNA level: *PLSCR4* was significantly downregulated in LipPD1 cells after 4, 8, and 12 days (normalized to *HPRT* and *TBP*; *n* = 4; *p* < 0.01). (**c**) Downregulation of PLSCR4 in APCs and LipPD1 cells during adipogenesis was observed at protein level (one representative experiment out of two is shown). *p*-values for panels (**a**,**b**) were determined via one-way ANOVA followed by post hoc Dunnett’s multiple comparison test (* *p* < 0.05, ** *p* < 0.01). PLSCR4, phospholipid scramblase 4; HPRT, hypoxanthine phosphoribosyl transferase; TBP, TATA-box binding protein; LipPD1, lipoma cell strain of a patient with PTEN hamartoma tumor syndrome (PHTS) carrying a heterozygous germline microdeletion in the PTEN gene.

**Figure 4 ijms-23-09787-f004:**
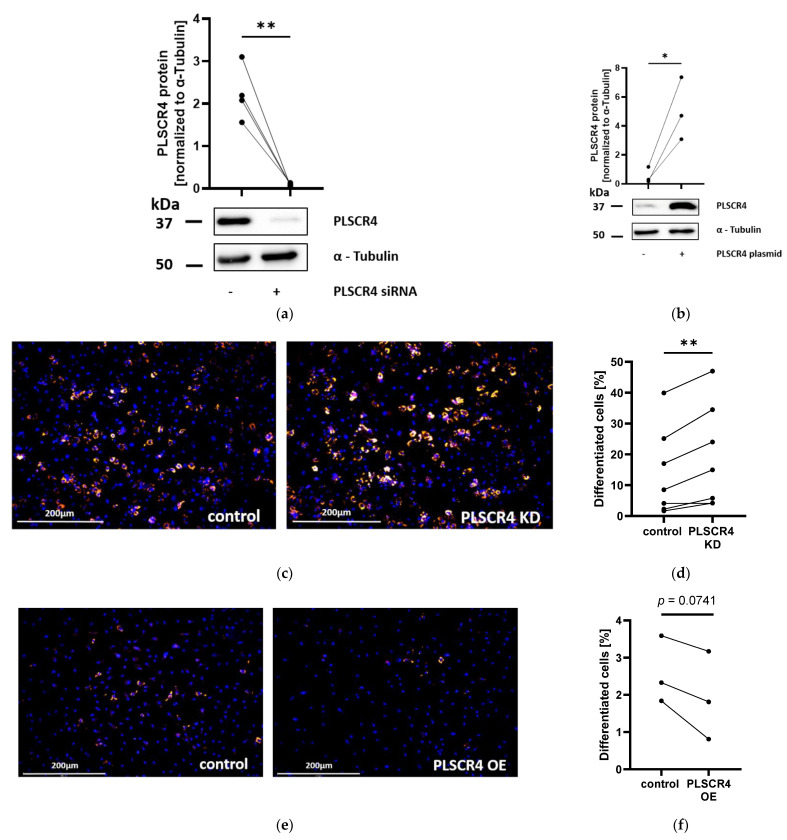
Effects of knockdown (KD) and overexpression (OE) of PLSCR4 on differentiation of APCs. (**a**) PLSCR4 KD: At protein level, PLSCR4 was reduced by 95.8% ± 15.0% (normalized to α-Tubulin; *n* = 4; *p* = 0.0077) in adipocyte progenitor cells. (**b**) PLSCR4 protein was upregulated 8.23 ± 1.79-fold (normalized to α-Tubulin; *n* = 3; *p* = 0.0446) in APCs transfected with a PLSCR4 expression plasmid. (**c**) Hoechst nuclear staining (blue) and Nile Red lipid staining (red) in control and PLSCR4-siRNA-transfected APCs after 8 days in adipogenic medium. (**d**) The differentiated cell fraction in PLSCR4 KD cells was increased 1.37 ± 0.09-fold (*n* = 7; *p* = 0.0056). (**e**) Hoechst nuclear staining (blue) and Nile Red lipid staining (red) in APCs with or without PLSCR4 OE after 8 days in adipogenic medium. (**f**) The fraction of differentiated cells decreased by 25.3% ± 7.3% (*n* = 3; *p* = 0.0741). Matched results were visualized via lines between data points (control versus PLSCR4 KD/OE). *p*-values were determined via paired *t*-test (* *p* < 0.05, ** *p* < 0.01). PLSCR4, phospholipid scramblase 4.

**Figure 5 ijms-23-09787-f005:**
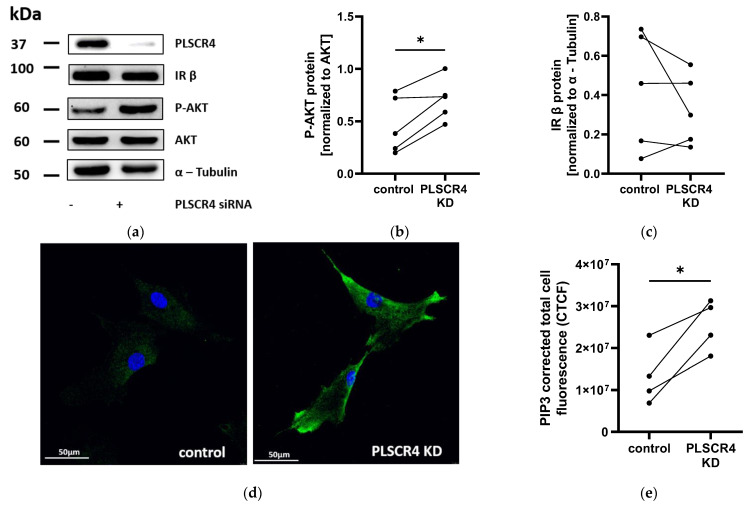
AKT phosphorylation and PIP3 immunofluorescence staining was increased in PLSCR4 knockdown (KD) adipocyte progenitor cells (APCs). (**a**) Western blots of PLSCR4, insulin receptor β subunit (IR β), and phosphorylated AKT (p-AKT T308) in PLSCR4 KD APCs, one representative image each. (**b**) AKT phosphorylation in PLSCR4 KD cells was increased by 52.0% ± 13.6% compared with control cells (phosphorylated AKT (p-AKT T308) normalized to total AKT protein; *n* = 5; *p* = 0.0186) (**c**) IR β subunit expression was variable between control cells and PLSCR4 KD cells and showed no regulation (IR β normalized to α-Tubulin; *n* = 5; *p* = 0.3283). (**d**) Hoechst nuclear staining (blue) and PIP3 immunofluorescence staining (green) in PLSCR4 KD and control APCs. (**e**) Analysis of corrected total cell fluorescence via ImageJ showed that PIP3 fluorescence was increased 1.92 ± 0.21-fold in PLSCR4 KD cells compared with controls (*n* = 4; *p* = 0.0226). Lines between individual data points indicate matched data from single experiments (control versus PLSCR4 KD). *p*-values for the experiments were determined via paired *t*-test (* *p* < 0.05). PLSCR4, phospholipid scramblase 4; IR β, Insulin receptor β subunit; AKT, serine/threonine kinase protein kinase B; p-AKT, phosphorylated AKT (T308).

## Data Availability

Not applicable.

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
