# Peer review of "Phospholipid Scramblase 4 (PLSCR4) Regulates Adipocyte Differentiation via PIP3-Mediated AKT Activation"

_ijms, 2022, doi:10.3390/ijms23179787_

Round 1
Reviewer 1 Report
This paper titled as "Phospholipid scramblase 4 (PLSCR4) regulates adipocyte differentiation via PIP3-mediated AKT activation" is interesting. In this study, the authors studying the mechanism of action of the under researched enzyme feels quite advantageous. At this stage, the manuscript does not meet requirement to be accepted for publication, it needs a improvement. Below are points that have to be addressed for improving it.
Comment 1. In the introduction part, I would like to add a description of the Phospholipid scramblase groups, and I think it would be good to mainly include the description of PLSCR4.
Comment 2. The english grammar of the overall sentences should be reaffirmed and careful revision is requied.
Comment 3. References should be again checked according to the instruction of the journal regulation.
Author Response
Answer to reviewer comments:
We would like to thank the reviewer for the valuable feedback and rapid review of our paper. Below you can find detailed responses to each comment. In order to answer the reviewers’ questions we conducted additional experiments, which we included in figure 3 c, supplementary figure S1 E and supplementary figure S4.
With the help of the reviewers we were able to substantially improve our manuscript. We attached a revised manuscript version with tracked changes.
Reviewer 1
This paper titled as "Phospholipid scramblase 4 (PLSCR4) regulates adipocyte differentiation via PIP3-mediated AKT activation" is interesting. In this study, the authors studying the mechanism of action of the under researched enzyme feels quite advantageous. At this stage, the manuscript does not meet requirement to be accepted for publication, it needs a improvement. Below are points that have to be addressed for improving it.
Thank you very much for your valuable comments. We revised the manuscript as advised.
Comment 1. In the introduction part, I would like to add a description of the Phospholipid scramblase groups, and I think it would be good to mainly include the description of PLSCR4.
Thank you for your helpful advice. We included the following passage in the introduction (page 2, line 67-70): Contrary to other transmembrane phospholipid transporters (flippases and floppases) scramblases are non-selective and ATP-independent bidirectional transporters. Their en-zyme activity depends on cellular calcium concentrations [10].”
Further, we added (page 2 line 81-84): “PLSCR5 is associated with brain function [12]. PLSCR4 was shown to be regulated in models of lipopolysaccharide induced acute respiratory distress syndrome [13], in sickle cell disease [14] and lipid and cholesterol metabolism [15]. However, little is known about PLSCR4 function in general and specifically in adipocytes.”
Comment 2. The English grammar of the overall sentences should be reaffirmed and careful revision is required.
Thank you for your comment. We revised our manuscript as advised.
Comment 3. References should be again checked according to the instruction of the journal regulation.
Thank you for your comment. We revised the references as advised.
Reviewer 2 Report
This article is very interesting, showing that PLSCR4 is involved in the regulation of adipogenesis by inhibiting the differentiation of adipocyte progenitor cells without affecting their proliferation by performing in vitro studies using human cells and in vivo studies using knockout mice. Authors also showed the effect of PLSCR4 was executed via PIP3/AKT-dependent, but IR-β/IRS-1-independent, pathways
This manuscript will be a candidate for the publication in IJMS if the following concerns are properly resolved.
Major concerns:
1)In Figure 3c, the data for “adipocyte progenitor cells” should be presented in addition to those for LipPD1 cells.
2) Detailed molecular mechanisms for PIP3/AKT-dependent signals to regulate adipocyte differentiation should be shown.
Minor concerns:
1) In Abstract, authors should add a brief and clear sentence that describes the unique involvement of Plscr4 compared to other Plscrs for readers to immediately understand the importance of Plscr4 in adipogenesis regulation.
Author Response
Answer to reviewer comments:
We would like to thank the reviewer for the valuable feedback and rapid review of our paper. Below you can find detailed responses to each comment. In order to answer the reviewers’ questions we conducted additional experiments, which we included in figure 3 c, supplementary figure S1 E and supplementary figure S4.
With the help of the reviewers we were able to substantially improve our manuscript. We attached a revised manuscript version with tracked changes.
Reviewer 2
This article is very interesting, showing that PLSCR4 is involved in the regulation of adipogenesis by inhibiting the differentiation of adipocyte progenitor cells without affecting their proliferation by performing in vitro studies using human cells and in vivo studies using knockout mice. Authors also showed the effect of PLSCR4 was executed via PIP3/AKT-dependent, but IR-β/IRS-1-independent, pathways.
This manuscript will be a candidate for the publication in IJMS if the following concerns are properly resolved.
Thank you very much for your valuable comments. We revised the manuscript as advised.
Major concerns:
- In Figure 3c, the data for “adipocyte progenitor cells” should be presented in addition to those for
LipPD1 cells.
Thank you for your suggestion, as advised we performed additional Western blot experiments for PLSCR4 in differentiated APCs. We observed similar results as for LipPD1 cells and as advised added the results to figure 3 c and supplement figure S1 E.
- Detailed molecular mechanisms for PIP3/AKT-dependent signals to regulate adipocyte differentiation should be
Thank you very much for this valuable comment. We and others could previously show that increased AKT activation will lead to phosphorylation of the transcriptional regulator FOXO1, which leads to its inactivation. As a result, FOXO1 cannot block transcription of the transcription factor SREBP1, which regulates the expression of lipogenic genes. To confirm this in the given setting, we performed additional Western blots for SREBP1 in PLSCR4 KD and control cells. We observed increased SREBP1 levels and added Figure S4 B and respective sections to results and discussion. Additionally we were interested whether the observed effect was specific to PIP3 and measured PIP2 and PIP3 species via HPLC-MS. Due to 40-fold lower abundance of PIP3 compared to PIP2, no reliable results could be obtained for PIP3. For PIP2 we detected a significant downregulation, suggesting that the observed effect is specific for the respective PIP species. While PIP3 levels were higher, PIP2 levels were lower in PLSCR4 KD. This is conclusive, since PIP2 and PIP3 are converted into each other via phosphorylation or dephosphorylation. We included results from PIP2 measurements in supplement figure S4 A and added paragraphs to the results and methods section.
We added (page 8 line 238-245): “In contrast, PIP2 levels, as determined via high performance liquid chromatography-mass spectrometry (HPLC-MS) measurement, were reduced by 23.1 ± 4.7% in PLSCR4 KD cells (n = 4, p = 0.0215) (Figure S4 A). We further investigated whether the phosphorylation of AKT leads to increased expression of the lipogenic transcription factor sterol regulatory element-binding protein 1 (SREBP1), which is known to be transcriptionally regulated by the AKT target FOXO1. SREBP1
protein was elevated in PLSCR4 KD cells by 28.9% ± 11.6% compared to control cells (n = 3, p = 0.116) (Figure S4 B).”
Discussion (page 9 line 279-281): “We observed increased PIP3, but decreased PIP2, in PLSCR4 KD APCs, which is in line with the observed AKT activation and adipogenesis.”
and (page 9 line 284-287): “Additionally, we observed increased expression of the lipogenic transcription factor SREBP1 in PLSCR4 KD cells, which is a downstream target of the PI3K pathway. Elevated SREBP1 levels may induce the observed effect on adipogenesis [5].”
Methods (page 12 line 412-443): 4.11. PIP2/PIP3 quantification by high performance liquid chromatography mass spectrometry (HPLC-MS)
Pellets from 106 cells were thawed on ice and 0.23 nmol of phosphatidylinositol 3,4-bisphosphate (17:0-20:4 PI (3,4)P2) and 0.21 nmol phosphatidylinositol 3,4,5-trisphosphate (17:0-20:4 PI (3,4,5)P3) (Avanti Polar Lipids, Alabaster, USA) were added to each sample before homogenization. Lipid extraction and derivatization were performed according to Clark et al. [29] with slight adaptations. In brief, 750 µl of ice cold MeOH/CHCl3/1 M HCl ((20/10/1, v/v/v)) was added to each sample, mixed thoroughly and incubated on ice for 5 min. Subsequently, 725 µl CHCl3 and 170 µl 2 M HCl were added, mixed thoroughly and incubated on ice for 10 min, followed by centrifugation to achieve phase separation (4°C, 5 min, 2000 × g). The lower phase was collected and re-extracted by addition of 708 µl MeOH/CHCl3/0.01 M HCl ((2/1/0.375, v/v/v)). Again, the lower phase was collected and 200 µl of MeOH were added for derivatization. Derivatization was performed using 50 µl of 2 M trimethylsilyldiazomethane in hexane. The components were allowed to react for 10 min at RT, before the reaction was stopped by addition of 6 µl glacial acetic acid. To induce phase separation 700 µl of MeOH/CHCl3/H2O ((2/1/0.375, v/v/v) were added to the mixture and mixed thoroughly (4°C, 5min, 2000 × g). The lower phase was collected and dried in vacuo. Extracts were resuspended in 25 µl isopropanol and briefly sonicated. 25 µl acetonitrile were added prior to chromatographic separation. The injection volume was 20 µl. Lipid extracts were separated by reversed phase liquid chromatography on a Ultimate 3000 RSLCnano system (Thermo Fisher Scientific, Germany) equipped with a Kinetex C18 column (Phenomenex, 2.6 µm 2.1×100 mm; 100 Å) by isocratic elution with solvent A (H2O containing 5 mM NH4HCO3 and 0.1% (v/v) formic acid) and B (acetonitrile/isopropanol, 50:50 (v/v) containing 0.1% (v/v) formic acid). Separation was performed at 35°C with a flow rate of 0.75 ml/min using a constant flow of 90% B. Mass spectra were recorded on an AB SCIEX QTRAP 5500 mass spectrometer (Sciex, Framingham, MA, USA) equipped with an ESI probe. Mass spectra were acquired in positive multi reaction monitoring (MRM) mode. MS parameters were set as follows: Spray Voltage 5.5 kV (ESI positive mode), Temp. 350°C, Curtain gas 42.0, Collision Gas 8.0, Ion Source Gas 30.0, Entrance Potential 10V, Declustering Potential 281 V for PIP2, Collision Energy 37 V, Excitation Energy 6 V for PIP2. Quantification was based on determination of area under curve values and normalized to the protein content of each sample.
Minor concerns:
1) In Abstract, authors should add a brief and clear sentence that describes the unique involvement of Plscr4 compared to other Plscrs for readers to immediately understand the importance of Plscr4 in adipogenesis regulation.
Thank you for this suggestion. We added a sentence to the abstract in accordance with the reviewer’s suggestion (page 1 line 27-29): “PLSCR4 is most highly expressed in human adipose tissue, compared to other phospholipid scramblases, suggesting a specific role for PLSCR4 in adipose tissue biology.”
Round 2
Reviewer 2 Report
Authors have made every effort to solve the issues. Therefore, the reviewer think that the revised manuscript has been sufficiently improved to warrant publication in IJMS.